# A micro RNA mediates shoot control of root branching

Moritz Sexauer [1,3,5], Hemal Bhasin [1,4,5], Maria Schön[1], Elena Roitsch[1,2], Caroline Wall[1], Ulrike Herzog[1] & Katharina Markmann [1,2,3] ✉

Plants extract mineral nutrients from the soil, or from interactions with mutualistic soil microbes via their root systems. Adapting root architecture to nutrient availability enables efficient resource utilization, particularly in patchy and dynamic environments. Root growth responses to soil nitrogen levels are shoot-mediated, but the identity of shoot-derived mobile signals regulating root growth responses has remained enigmatic. Here we show that a shoot-derived micro RNA, miR2111, systemically steers lateral root initiation and nitrogen responsiveness through its root target *TML* (*TOO MUCH LOVE*) in the legume *Lotus japonicus*, where miR2111 and *TML* were previously shown to regulate symbiotic infections with nitrogen fixing bacteria. Intriguingly, systemic control of lateral root initiation by miR2111 and *TML/HOLT* (*HOMOLOGUE OF LEGUME TML*) was conserved in the nonsymbiotic ruderal *Arabidopsis thaliana*, which follows a distinct ecological strategy. Thus, the miR2111-*TML/HOLT* regulon emerges as an essential, conserved factor in adaptive shoot control of root architecture in dicots.

Root systems are the main contact point of land plants with soluble nutrients. Adapting the root surface area to nutrient availability in the substrate is thus a key aspect of endogenous resource management in land plants.

Consistently, the perception of both restrictive and sufficient levels of nitrate, a frequently limiting macronutrient, induces root architectural adaptations. To this end, plants can enhance lateral root growth, and thus root surface area, either under deficient nitrate conditions or within local, nitrogen-rich patches. This process is termed foraging[1]. Within a given root segment, several factors have been suggested to be involved in regulating nitrate foraging locally. In *Arabidopsis thaliana* (Arabidopsis), the nitrate transceptor protein NRT1.1 controls biosynthesis and transport of auxin, thereby mediating local repression of lateral root development where perceived nitrate levels are low[2]. Downstream of *NRT1.1*, the GRAS transcription factor NIN LIKE PROTEIN 7 (NLP7) was shown to induce expression of the MADS-box gene *ANR1*, mediating further transcriptional changes that specifically

promote lateral root elongation in nitrate rich soil patches[3]. Signalling via CLE (CLAVATA3/ESR) peptides and the leucine-rich repeat receptor kinase CLAVATA1 (CLV1) was further shown to be involved in nitrate-dependent local regulation of lateral root emergence in Arabidopsis[4]. A similar role has been assigned to the putative CLV1-ortholog *HYPERNODULATION ABERRANT ROOT FORMATION1* (*HAR1*) in *Lotus japonicus* (Lotus)[5]. Balancing need and availability of nutrients is a challenge concerning the plant as a whole. Adaptations to nutrient stress thus require communication not only within, but also across plant organs, suggesting that they involve systemic signalling circuits linking above- and belowground tissues. Grafting experiments demonstrated that both shoot and root expression of *HAR1* is required for nitrate-dependent adaptation of lateral root growth in Lotus[5]. This suggests a dual root-specific as well as systemic role of the CLE-HAR1 signalling node. Consistently, Lotus CLE-RS peptides were shown to be competent of xylem-based root-shoot mobility following arabinosylation, and can directly bind to HAR1[6]. In addition, both C-terminally encoded peptide

[1]Eberhard-Karls-University, Centre for Molecular Biology of Plants, Tübingen, Germany. [2]Martin-Luther-University Halle-Wittenberg, Institute for Genetics, Halle/Saale, Germany. [3]Present address: Julius-Maximilians-University, Julius-von-Sachs Institute for Biosciences, Würzburg, Germany. [4]Present address: University of Toronto – Scarborough, Department of Biological Sciences, Toronto, ON, Canada. [5]These authors contributed equally: Moritz Sexauer, Hemal Bhasin. ✉e-mail: katharina.markmann@uni-wuerzburg.de

(CEP) hormones[1,2] as well as cytokinins[3,4] act as systemic root-shoot factors signalling low or high root nitrate content, respectively. In Arabidopsis, upon CEP perception by the CEP receptors CEPR1/2, shoot-produced CEP Downstream (CEPD) and CEPD LIKE (CEPDL) peptides translocate to roots to regulate nitrate uptake via transcriptional as well as post-translational regulation of *NRT2.1*[7,8]. In the legume *Medicago truncatula* (Medicago), the putative CEPR1 orthologue *COMPACT ROOT ARCHITECTURE 2* (*CRA2*) similarly steers *NRT2.1* dependent nitrate uptake by mediating *CEP1* dependent expression regulation[9,10].

While shoot-root mobile CEPD and CEPDL signals systemically regulate root nitrate uptake, systemic shoot factors mediating nitrate-dependent root growth adaptations are so far unknown. We previously observed that a shoot-derived, phloem-mobile micro RNA, miR2111, regulates the formation of symbiotic infections and nitrogen-fixing nodule organs in *Lotus japonicus* (Lotus) roots inoculated with rhizobial bacteria[11]. miR2111 post-transcriptionally targets the root-expressed F-Box Kelch-repeat gene *TOO MUCH LOVE* (*TML*), which represses symbiosis[11,12]. Both CLE-RS/HAR1[11] and CEP/CRA2 signalling nodes[12] regulate miR2111 abundance. Shoot miR2111 accumulation is repressed in the presence of sufficient nitrate as well as of compatible rhizobia[11], releasing *TML* mRNA from posttranscriptional regulation and restricting symbiosis progression. Interestingly, miR2111 and *TML* are not restricted to plants establishing root nodule symbiosis, but are conserved across dicot lineages. The Arabidopsis genome contains two *MIR2111* precursor gene loci[13], both encoding a single miR2111 isoform that specifically targets the F-box Kelch-repeat gene At3g27150[13], a *TML* homolog[14] of unknown function. In comparison, the Lotus genome contains seven *MIR2111* loci encoding three different isoforms[11,15]. On this basis, we hypothesized that miR2111 may have a conserved role in regulating lateral organ formation in roots also in nonsymbiotic settings, and have undergone functional diversification in nodulating lineages. Our work identifies miR2111 as a missing link signalling shoot nitrogen status to root organs and regulating adaptive root growth responses in a nitrate-dependent manner.

## Results and discussion
### miR2111 is a shoot factor regulating root architecture
To investigate possible conserved, symbiosis-independent functions of miR2111 in root system architecture control, we analysed root

systems of plants mis-expressing the miRNA. Indeed, Lotus plants expressing a *pUBQ1::MIR2111-3* transgene resulting in overabundance of mature miR2111 (Fig. 1a) generated less lateral roots than wild type plants (Fig. 1b). miR2111 is produced primarily in shoots, and is proposed to translocate to roots via the phloem[11,15]. Phloem-mobile miR-NAs were recently suggested to translocate as fully processed duplices, rather than as pri- or pre-miRNA precursors[16]. Consistently, we could trace plant specific mature miR2111 transcripts in aphids (*Planococcus citri*) feeding on Lotus, indicating its presence in the phloem sap (Supplementary Fig. 1a–c). To investigate whether shoot-derived miR2111 is indeed functional in Lotus roots and sufficient to regulate lateral root number, we grafted *pUBQ1::MIR2111-3* expressing shoots onto wild type root stocks (Fig. 1c). Roots of chimeric plants showed enhanced levels of miR2111 (Fig. 1d), and fewer emerged lateral roots compared to control grafts (Fig. 1c, e), confirming that shoot miR2111 indeed translocates to roots to steer lateral root numbers.

### miR2111 regulates lateral root initiation through its target *TML*
miR2111 was proposed to directly target *TML* for posttranscriptional regulation in Lotus as well as Medicago[11,12]. Consistently, roots of *pUBQ1::MIR2111-3*/wild type (shoot/root) grafts had significantly lower *TML* levels than wild type/wild type controls (Fig. 1f). *tml* knockout mutants developed less lateral roots than wild type plants (Fig. 2a, b), and were phenotypically indistinguishable from *pUBQ1::MIR2111-3* plants (Fig. 2a, b), suggesting that *TML* is the main target of miR2111 activity in lateral root control. Interestingly, this was equally the case when all initiated lateral roots, including pre-emerged root primordia as well as emerged lateral roots, were considered (Fig. 2c, Supplementary Fig. 2). Lateral root initiations were also reduced in grafted plants expressing *pUBQ1::MIR2111-3* in their shoots compared to wild type/wild type control grafts (Fig. 2d). Using Crispr-CAS9 technology, we generated line *mir2111-3-1*, which possesses a 12 bp deletion in the stem-loop region of the *MIR2111-3* locus in immediate proximity to the mature miRNA2111a sequence (Supplementary Fig. 3a–c). *mir2111-3-1* plants showed significantly lower miR2111 abundance than wild type plants (Supplementary Fig. 4a) and, consistently, higher *TML* transcript levels (Supplementary Fig. 4b). In line with the observed reduced primordium formation in miR2111 overexpressors (Fig. 2c, d), lateral root

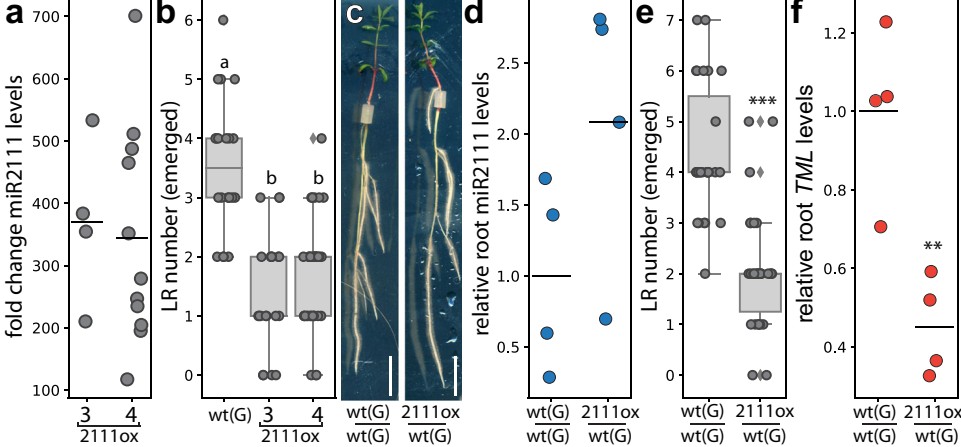

**Fig. 1 | Shoot-derived miR2111 regulates lateral root (LR) numbers in *L. japonicus* (Lotus). a** miR2111 abundance fold change compared to Gifu wild type (wt(G)) plants, and (**b**) LR count in transgenic *pUBQ1::MIR2111-3* (2111ox) expressing lines (#3, 4) compared to wt(G). Line #3 was used for further analysis. **c–f** 2111ox / wt(G) (shoot / root) grafts compared to wt(G) / wt(G) control grafts. **c** Example of grafted plants. Scale bars equal 1 cm. **d** miR2111 levels, (**e**) LR numbers and (**f**) *TML* levels in roots of respective grafts. **a, d, f** qRT-PCR analyses. RNA levels are relative to those of two reference genes. RNA was extracted from root (**d, f**) or shoot samples (**a**). **a** Adult plants grown in soil. **b–g** Plants were grown at 0 mM nitrate and evaluated

or harvested after two weeks of cultivation. Student's *t*-test (**p ≤ 0.01; ***p ≤ 0.001) (**d–f**) or analysis of variance (ANOVA) and post-hoc Tukey testing (p ≤ 0.05) (**b**), with distinct letters indicating significant differences. All experiments were in ecotype Gifu B-129 (wt(G)). Sample size, replicates and exact p-values are listed in the Source Data file. Dotplots show individual data points and a line indicating their average value. Boxplot central line shows median value, box limits indicate the 25th and 75th percentile. Whiskers extend 1.5 times the interquartile range, or to the last datapoint. Individual datapoints are represented by dots.

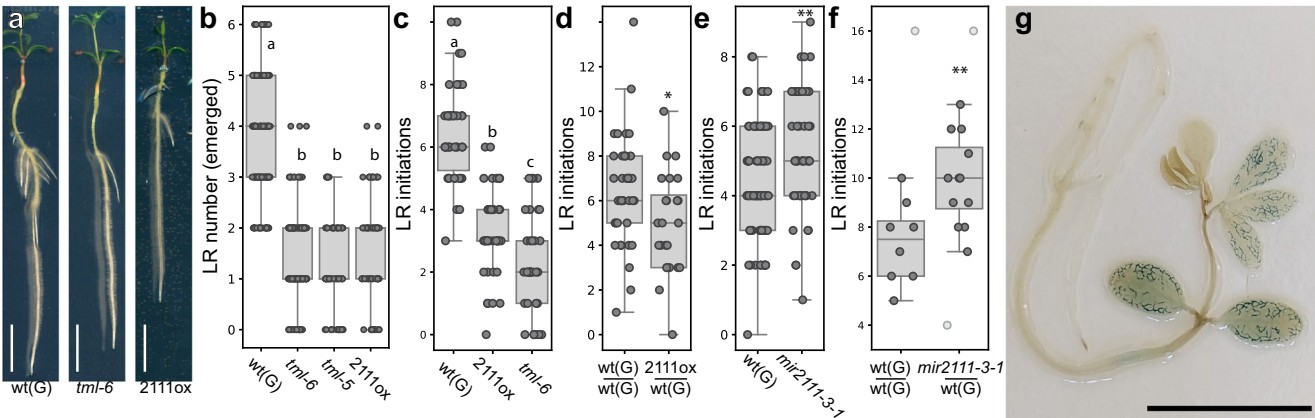

**Fig. 2 | *TML* mediates miR2111 control of *L. japonicus* (Lotus) lateral root (LR) initiation. a** Root phenotype of Gifu wild type (wt(G)), *tml-6* and *pUBQ1::MIR2111-3* plants (2111ox). Scale bars equal 1 cm. **b** Emerged LR numbers in wt, *tml-6*, *tml-5* and 2111ox plants. **c–e** Number of LR initiations (emerged and primordial stages combined) in wt(G), 2111ox and *tml-6* plants (**c**), on 2111ox / wt(G) (shoot / root) grafts compared to wt(G) / wt(G) control grafts (**d**), in *mir2111-3-1* compared to wt(G) plants (**e**) and on *mir2111-3-1* / wt(G) (shoot / root) grafts compared to wt(G) / wt(G) control grafts (**f**). **f** light grey dots represent data points not considered in the statistical analysis due to strong divergence of primary root length in the respective plants from the mean. **g** Plants expressing *pMIR2111-3:GUS* show pronounced GUS activity in leaf veins, while roots are free of visually traceable activity. Scale bar

equals 1 cm. A total of 30 tested plants showed a similar expression pattern. **b** Datapoints are identical to datapoints at 0 mM nitrate in Fig. 3a. **a–e** Plants grown at 0 mM nitrate. Comparisons used Student's *t*-test (*$p \le 0.05$; **$p \le 0.01$) (**d**, **e**) or analysis of variance (ANOVA) and post-hoc Tukey testing ($p \le 0.05$) (**b**, **c**), with distinct letters indicating significant differences. All experiments were in ecotype Gifu B-129 (wt(G)). **a–g** Plants were evaluated or harvested after two weeks of cultivation (**a–c**, **e** & **g**) or grafting (**d**, **f**). Sample size, replicates and exact *p*-values are listed in the Source Data file. Boxplot central line shows median value, box limits indicate the 25th and 75th percentile. Whiskers extend 1.5 times the interquartile range, or to the last datapoint. Individual datapoints are represented by dots.

initiation numbers were higher in *mir2111-3-1* plants compared to wild type plants (Fig. 2e), and grafts of *mir2111-3-1* shoots on wild type root stocks equally showed an enhanced lateral root initiation compared to control grafts. This is consistent with a shoot specific expression pattern of the *MIR2111-3* locus (Fig. 2f, g), and confirms that shoot miR2111 is required for lateral root initiation control. Taken together, these data suggest that shoot-derived miR2111 is both sufficient and necessary for modulating lateral root initiations via *TML* (Fig. 2c–e, Supplementary Fig. 3a–c). Interestingly, this is in addition to the described function of shoot-derived miR2111 in systemic nodule number control in the context of symbiosis autoregulation (Supplementary Fig. 4c)[11,15].

## The miR2111/*TML* regulon controls root branching in response to nitrate levels

Since root nodulation symbiosis, a known activity context of miR2111-*TML*, is an adaptation to nitrogen limitation, we hypothesized that this regulon may also help adapting root architecture to nitrogen availability.

Lotus showed enhanced lateral root numbers under nitrogen starvation (Fig. 3a). This was ecotype-independent (Fig. 3a, b) and is consistent with the nitrate foraging responses reported in other plants[17]. Notably, this trend was only apparent for emerged lateral roots. Lateral root primordia, on the contrary, were more abundant under nitrate sufficient conditions compared to deficiency. This results in a positively nitrate-correlated (Fig. 3c, d) or nitrate-independent (Supplementary Fig. 5a) sum of initiated roots.

Following a likewise trend, mature miR2111 levels were negatively correlated with nitrate availability in a dosage-dependent manner in both shoots and roots (Fig. 3e, f), indicating an involvement in systemic nitrogen response signalling. The levels of *TML* transcripts, which were only detected in roots, showed a complementary, inverse pattern (Fig. 3g), suggesting *TML* suppression by systemic miR2111 under nitrogen starvation conditions. Indicating ecotype specific differences, this pattern was particularly apparent in the Lotus ecotype MG-20, consistent with a more pronounced responsiveness of lateral root initiations to nitrate availability compared to Gifu B-129 (Supplementary Fig. 5a–d).

Compared to wild type plants, both miR2111 overexpressors and *tml* mutants had a consistently lower number of lateral roots, and emerged lateral root number was independent of nitrate availability (Fig. 3a, b). The same was true for lateral root initiation, not only in the ecotype Gifu B-129, but also in MG-20, where a nitrate-dependent increase in lateral root primordia is more strongly pronounced than in Gifu B-129 (Fig. 3c, d and Supplementary Fig. 5a). Shoot specific overexpression of *MIR2111-3* using heterografting experiments induced a loss of nitrate responsive lateral root initiation in chimeric plants with MG-20 root stocks (Supplementary Fig. 6a, b), suggesting that shoot-derived miR2111 efficiently represses this response. On this basis, we predicted that nitrate-independent *TML* transcript levels in Gifu B-129 (Supplementary Fig. 5d) may prevent adaptive primordia formation in this ecotype. Indeed, increased *TML* transcript levels in *mir2111-3-1* compared to wild type plants (Supplementary Fig. 4b) resulted in a positive nitrate response of lateral root initiation numbers in the ecotype Gifu B-129 as well (Supplementary Fig. 6c), indicating that miR2111 mediated *TML* control is necessary for the ecotype-specific attenuation of root system response to nitrate observed in Gifu B-129. The combined phenotypic and molecular data suggests a role of the miR2111-*TML* regulon in lateral root initiation and adaptive emergence in response to nitrate, with miR2111 systemically repressing *TML*. Contrasting with their respective roles in symbiosis, our data identify miR2111, a positive regulator of nodule numbers, as a repressor of root primordia, and *TML* as a root primordial activator (Fig. 3h). The data further reveal that nitrate dependent regulation of primordia emergence into full lateral roots strictly requires the presence of functional *TML* (Fig. 3a–c), but does not correlate with *TML* transcript abundance (Fig. 3g, h, Supplementary Fig. 5d). This suggests involvement of additional factors in regulating nitrate responsive emergence of *TML*-dependent lateral root primordia.

Nitrate perception and nitrogen starvation have been found to trigger local and systemic responses involving physiological and morphological adaptations[17]. We thus performed split root assays to identify the trigger underlying miR2111 regulation under asymbiotic conditions. Roots growing on nitrogen starvation medium contained low miR2111 levels if other roots of the same plant experienced nitrate sufficiency (Fig. 3i). This suggests that miR2111 accumulation is not

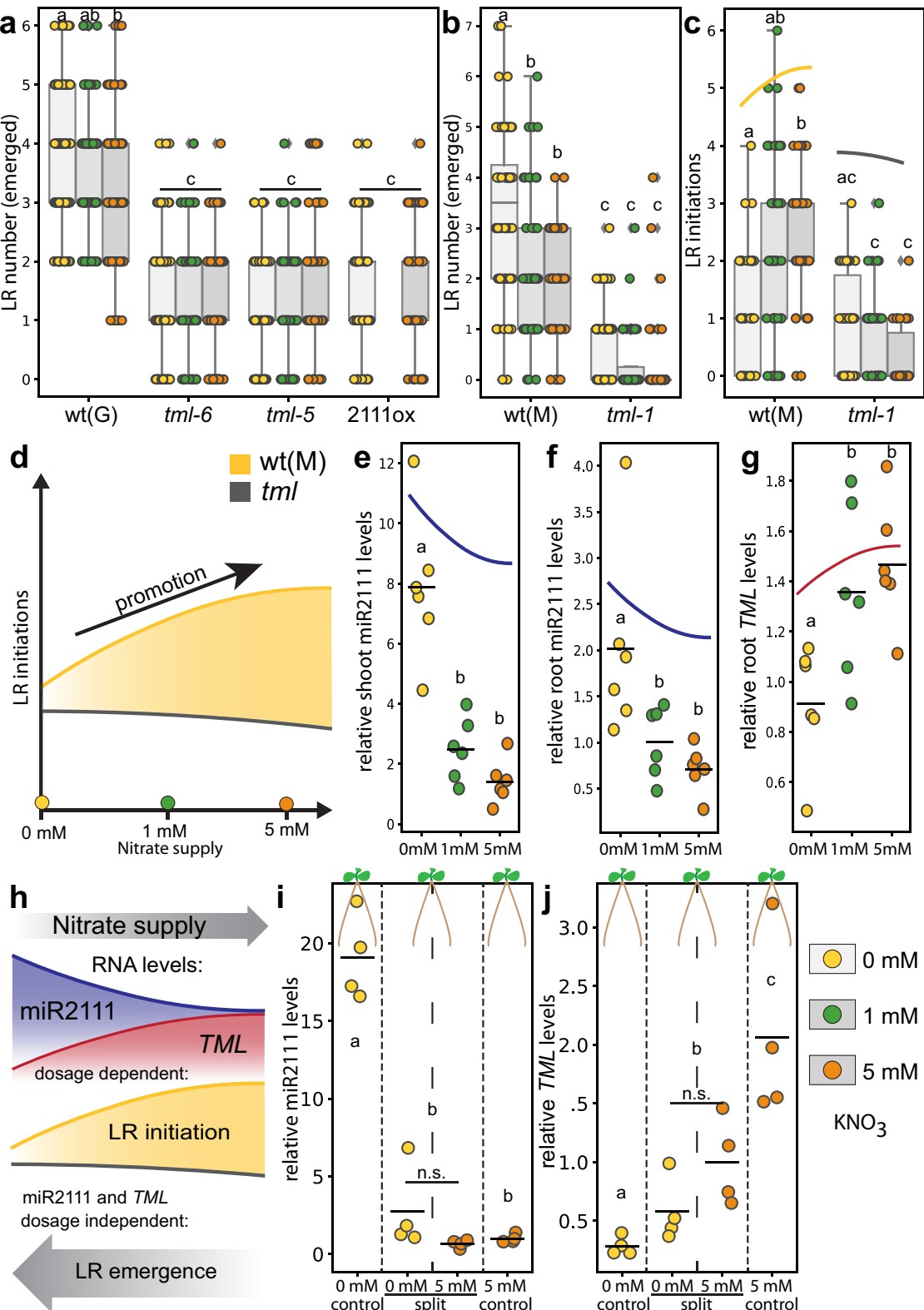

triggered by roots experiencing nitrate starvation, but rather is systemically repressed by roots exposed to nitrate sufficiency (Fig. 3i), implying that miR2111 levels are regulated through nitrate supply rather than deficiency. *TML* levels in these roots were complementary yet intermediate (Fig. 3j). Consistent with previous observations, *TML* abundance is thus likely subject to additional regulatory factors[11].

Apart from nitrate, rhizobial infection triggers changes in miR2111 and *TML* transcript abundance (ref. [11]; Supplementary Fig. 7 a–c), and

miR2111 acts as a positive regulator of nodule organogenesis by repressing the nodulation inhibitor *TML*[11]. We thus wondered how miR2111-*TML* dynamics affect lateral root formation under symbiotic conditions. Interestingly, in both wild type and *tml-6* mutant plants, symbiotic infection led to a decrease of lateral root initiations (Supplementary Fig. 7d), implying that an additional *TML*-independent regulation of lateral root initiation overlays miR2111-*TML* dependent primordium control under symbiotic conditions.

**Fig. 3 | Systemic N status controls lateral root (LR) initiations via the miR2111-**
**_TML_ regulon in _L. japonicus_ (Lotus). a, b** Emerged LRs in (**a**) Gifu B-129 wild type
(wt(G)), _tml-6_, _tml-5_ and _pUBQ1::MIR2111-3_ (2111ox), and in (**b**) MG20 wildtype
(wt(M)) and _tml-1_ plants. **c** Number of LR initiations (emerged plus primordial
stages) in wt(M) and _tml-1_ plants. **d** Simplified model of nitrate dependency of LR
initiations in wt and _tml_ mutant plants. **e, f** Relative mature miR2111 levels in shoots
(**e**) and roots (**f**). **g** Relative _TML_ levels in same wild type root systems as in (**f**).
**h** Simplified model outlining nitrate dependency of miR2111 and _TML_ levels, and
root architectural responses. **i, j** Split root experiments. Relative miR2111 (**i**) and
_TML_ (**j**) levels in secondary roots of wt(M) plants. **e–g, i, j** qRT-PCR analyses. RNA
levels are relative to those of two reference genes. **a, b, e–g, i, j** Tissue harvest /

analysis after two weeks and (**c**) 10 days of cultivation. Comparisons used analysis
of variance (ANOVA) and post-hoc Tukey. testing ($p \leq 0.05$), with distinct letters
indicating significant differences and additional Student's _t_-test (**i, j**) comparing
only the split roots (n.s. $p > 0.05$). **c–h** Trendlines are simplified and not to scale.
Plants grown at indicated nitrate concentrations. Sample size, replicates and exact
p-values are listed in the Source Data file. Dotplots show individual data points and
a line indicating their average value. Boxplot central line shows median value, box
limits indicate the 25th and 75th percentile. Whiskers extend 1.5 times the inter-
quartile range, or to the last datapoint. Individual datapoints are represented
by dots.

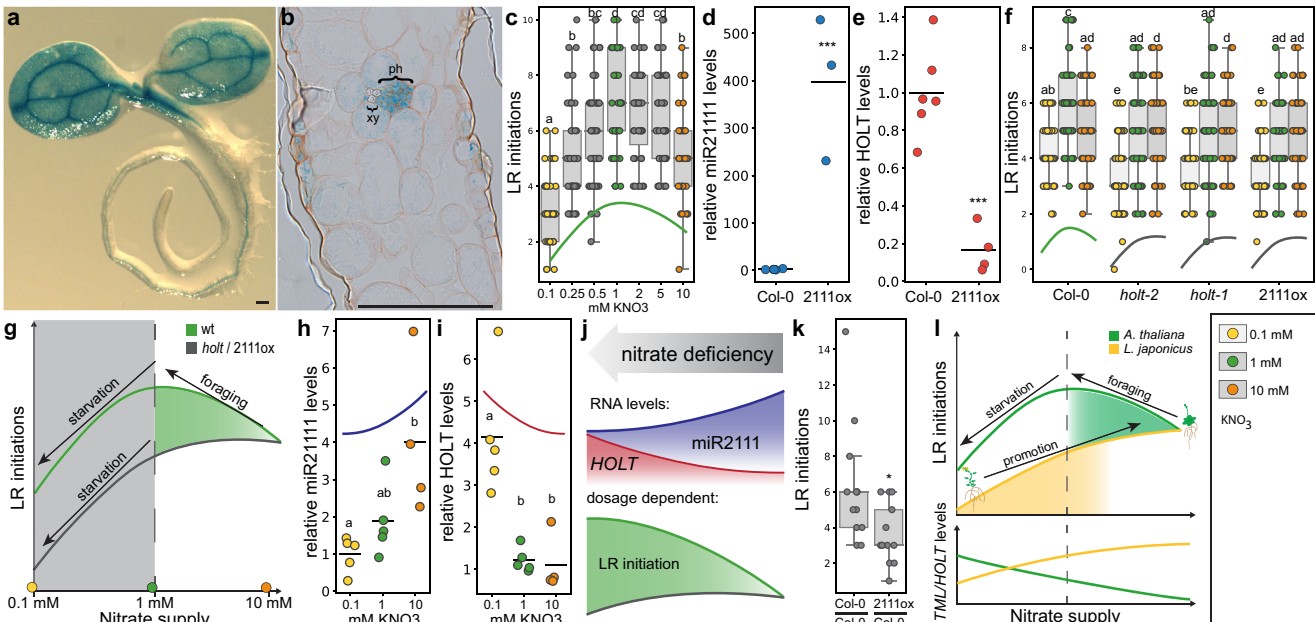

**Fig. 4 | The _A. thaliana_ (Arabidopsis) miR2111-_HOLT_ regulon controls lateral**
**root (LR) initiation at moderate nitrate starvation. a, b** Stably transformed _A.
thaliana_ (Arabidopsis) plants expressing _pMIR2111b:GUS_ show predominant GUS
activity in the phloem of leaf veins. **b** Leaf cross section of _pMIR2111b:GUS_ plants.
ph, phloem and xy, xylem. **c** Numbers of LR initiations at different nitrate con-
centration in Arabidopsis wild type (Col-0) plants. **d, e** miR2111 (**d**) and _HOLT_ (**e**)
levels in Col-0 and _p35s::MIR2111b_ expressing plants (2111ox) at 1 mM nitrate. **f** LR
initiations in _holt-1_, _holt-2_ and 2111ox plants compared to Col-0. **g** Schematic model
of nitrate responsiveness of Arabidopsis lateral root initiations. _holt-1_ and 2111ox
plants lack a foraging response at moderate nitrate starvation. **h, i** miR2111 (**h**) and
_HOLT_ (**i**) levels of Col-0 at varying nitrate concentrations. **j** Simplified model out-
lining nitrate dependency of miR2111 and _HOLT_ levels, and of root architectural
responses in Arabidopsis between 1 and 10 mM nitrate. **k** Number of LR initiations
on 2111ox / Col-0 (shoot / root) grafts compared to Col-0 / Col-0 control grafts at
1 mM $KNO_3$. **l** Combined simplified model of Arabidopsis and _L. japonicus_ (Lotus)
root responses to varying nitrate supply. _HOLT_ positively correlates with LR

initiations in both species, but nitrate dependent abundance patterns of both LR
initiations and _TML/HOLT_ levels are opposite. **d, e, h, i** qRT-PCR analyses. RNA levels
are relative to those of two reference genes, whole plant tissue harvested 10 days
after germination. **c, f, k** Analysis seven days after germination (**c, f**) or after graft
regeneration (**k**). Comparisons used analysis of variance (ANOVA) and post-hoc
Tukey testing ($p \leq 0.05$), with distinct letters indicating significant differences
(**c, f, h, i**) or Student's _t_-test (*$p \leq 0.05$) (**d, e, k**). **a, b** Scale bars equal 200 μm (**a**) or
100 μm (**b**). **a, b** all 21 tested plants of 3 independent lines showed a similar
expression pattern. Analysis of three independent lines showed similar results.
**c, f, h, i** Plants were grown at indicated nitrate concentrations using ½ strength MS
media free of other nitrogen sources. **c, f–j, l** Trendlines are simplified and not to
scale. Sample size, replicates and exact _p_-values are listed in the Source Data file.
Dotplots show individual data points and a line indicating their average value.
Boxplot central line shows median value, box limits indicate the 25th and 75th
percentile. Whiskers extend 1.5 times the interquartile range, or to the last data-
point. Individual datapoints are represented by dots.

## Systemic root control by miR2111/_TML_ is phylogenetically conserved

Root architecture adaptation to abiotic stimuli is an ancient necessity
and a core developmental feature of land plants that is phylogeneti-
cally widespread[17] and thus precedes the evolution of nitrogen-fixing
nodulation symbiosis. Consistently, the miR2111-_TML_ regulon is con-
served in non-nodulating plants, including the nonsymbiotic plant
_Arabidopsis thaliana_ (Arabidopsis)[11,14]. Arabidopsis possesses two
_MIR2111_ precursor loci generating a single isoform identical to
_Lj_miR2111a (Supplementary Fig. 8a, b)[11]. Phylogenetic analysis revealed
one putative _TML_ orthologue, which we named _HOMOLOGUE OF
LEGUME TML (HOLT)_, featuring a miR2111 complementary site in the
coding sequence[11,14] (Supplementary Fig. 9a, b). Consistent with the

expression pattern of _MIR2111_ loci in Lotus, Arabidopsis
_pMIR2111a/b:GUS_ expressing lines showed predominant GUS activity in
leaf vein phloem cells (Fig. 4a, b and Supplementary Fig. 10a, b), sug-
gesting systemic mobility[18]. These observations are in line with organ
specific expression data[19] (Supplementary Table 1). Similar to what has
been previously observed for selected Lotus _MIR2111_ precursor
genes[15], moderate _pMIR2111a/b:GUS_ activity was evident in mature root
parts of Arabidopsis as well (Fig. 4a and Supplementary Fig. 10a),
although _MIR2111a/b_ precursor transcripts were not traceable in pub-
licly available RNAseq datasets[19] (Supplementary Table 1). Like in
Lotus, Arabidopsis lateral root initiation numbers depend on nitrogen
supply, peaking around 1 mM nitrate under long day conditions in
plate-grown wild type plants (Fig. 4c). In comparison, plants grown

under starvation or saturating conditions show reduced numbers of lateral root initiations (Fig. 4c). The observed increase of lateral root numbers under moderately deficient (1 mM nitrate) as compared to sufficient (10 mM nitrate) nitrogen supply has previously been associated with nitrogen dependent root architectural adaptations commonly referred to as foraging response[1]. To evaluate a possible role of the miR2111/HOLT regulon in nitrogen foraging related lateral root initiation in Arabidopsis, we generated transgenic lines overexpressing miR2111 under the control of a Cauliflower Mosaic Virus 35 s promoter fragment, showing a concomitant reduction in HOLT levels (Supplementary Fig. 11a, b). All tested lines showed reduced lateral root initiation compared to wild type plants in the T2 generation (Supplementary Fig. 11c). We chose a representative line, #3, for further propagation, as it showed stable overabundance of miR2111 and a corresponding reduction of TML transcript abundance in the T3 generation (Fig. 4d, e). We further isolated Arabidopsis holt-1 and holt-2 mutants lacking a traceable full-length HOLT transcript (Supplementary Fig. 12a, b). holt-1, holt-2 and p35s::MIR2111b plants showed significantly reduced lateral root initiations at low and moderate nitrate concentrations compared to wild type plants (Fig. 4f). Notably, they failed to show a traceable foraging response (Fig. 4f,g). Wild type plants exposed to severe nitrogen limitation repress lateral root development, a response known as a survival strategy[1], which is thought to involve the nitrate transporter NRT1.1[20] as well as locally induced lateral root inhibition through the CLAVATA3/CLAVATA1 signalling module[4]. Transcript abundance of NRT1.1 and other NRTs was not significantly altered in holt-1 or p35s::MIR2111b as compared to wild type plants (Supplementary Fig. 13a–d). Consistent with a HOLT independent mechanism, a successive reduction in lateral root initiation numbers at nitrate levels <1 mM was retained in holt-1, holt-2 and p35s::MIR2111b in a similar way as in wild type plants (Fig. 4f, g). In wild type plants, miR2111 levels correlate positively with nitrate concentration (Fig. 4h), consistent with low HOLT levels at high nitrate supply (Fig. 4i). The integration of phenotypic and molecular data reveals that, in line with observations in Lotus, HOLT levels positively correlate with lateral root initiations (Fig. 4j). To investigate whether shoot-derived miR2111 is sufficient to regulate root architecture in Arabidopsis as observed in Lotus, we analysed p35s::MIR2111b/Col-0 (shoot/root) grafts. These had significantly less lateral root initiations than Col-0/Col-0 control grafts (Fig. 4k), confirming miR2111 as a systemically acting, mobile regulator of lateral root initiation across dicot plant lineages.

In line with divergent habitat requirements and ecological strategies of the symbiotic Lotus[21,22] and the asymbiotic ruderal Arabidopsis[23], abundance patterns of lateral root primordia with respect to external nitrogen supply were distinct in these two species (Fig. 4l). Yet, consistent with a conserved positive role of TML/HOLT in nitrate-dependent lateral root initiation, TML/HOLT RNA levels were upregulated in both species under nitrate conditions triggering abundant lateral root primordia. Accordingly, in either species, miR2111 levels were low under such conditions, in line with a negative effect on TML/HOLT levels and lateral root initiation. The dynamic response pattern of the Arabidopsis root system reflected in integrating distinct and functionally overlapping regulatory nodes (refs. 4,20, this study) indicates its capacity to populate a wide variety of soils[23]. Our data suggests that Lotus, as a pioneer lineage that is primarily competitive on nitrogen poor soils[24], initiates additional root primordia under starvation conditions that have a limiting effect on Arabidopsis root architecture (Fig. 4l). The lack of a strong nitrogen starvation response in Lotus could be explained by the formation of nitrogen fixing symbiosis, which prevents nitrogen starvation even on nitrogen poor soils.

An important role of the miR2111-TML/HOLT regulon in adapting plant root systems to their natural habitat is in line with the observed differences in lateral root abundance patterns between Lotus MG-20 and Gifu ecotypes (Fig. 3a–c, Supplementary Fig. 5a, Supplementary Fig. 6a–c). Lotus japonicus underwent intense diversification during evolution and encompasses more than 130 ecotypes that have adapted to a wide range of environmental conditions on the Japanese Islands[21]. A time course experiment revealed an increasing difference between wild type and tml-1 in lateral root numbers over time. Here, tml-1 plants showed significantly less biomass production in both below- and aboveground tissues compared to wild type (Supplementary Fig. 14a, b), suggesting that root architecture adaption plays an important role in plant productivity and fitness. We have no evidence for a direct involvement of the miR2111-TML/HOLT regulon in nitrate uptake, and mRNA levels of nitrate transporter genes NRT1.1, NRT1.5, NRT2.1 and NRT3.1 are unaltered in holt-1 and p35s::MIR2111b lines compared to wild type controls (Supplementary Fig. 13). This is in contrast to CEP/CEPD/CEPDL2 mediated regulation of nitrate uptake via NRT2.1 regulation[7,8], and suggests an indirect role of the miR2111-TML/HOLT regulon in nutrient uptake regulation by altering the extent of the root surface area (Supplementary Fig. 13).

Alteration of the root depletion zone also affects the uptake of other nutrients. Interestingly, in Arabidopsis, miR2111 was shown to be induced by phosphate starvation[25], and Arabidopsis is known to adapt its root architecture to phosphate availability[26]. This could hint to a more general role of miR2111 in adapting root architecture to nutrient availability.

The presented data identify miR2111 and TML/HOLT as conserved factors in root architectural control, suggesting that they were evolutionarily co-opted by rhizobial nodulation symbiosis to regulate root responses to symbiotic bacteria, and organogenesis of nodule organs[11]. Consistent with this hypothesis, the transcription factors SCARECROW and SHORTROOT[27], as well as ASYMMETRIC LEAVES2-LIKE18[28,29] and STYLISH[30] mediating auxin signalling hold dual roles in nodule organogenesis and root development, and comparative transcriptome analysis of lateral root and nodule primordia further supports generic ties between these organs[28].

Our data reveal the miR2111-TML/HOLT regulon as a key factor in systemic control of root system architecture and lateral root organ number. An exciting future challenge will be determining the molecular activity of the TML/HOLT protein, a proposed component of the E3 Ubiquitin ligase complex in Arabidopsis[31] with a possible role in mediating degradation of target transcription factors[31]. Determining downstream effectors will help us better understand how the miR2111-TML/HOLT regulon functionally integrates with hormonal networks and other regulators of root growth.

## Methods
### Plant and bacterial resources
Plants for root architecture analyses, qPCR assays and GUS stainings were Lotus japonicus L. ecotype Gifu B-129 (wild type, tml-5 (line ID 30013998), tml-6 (line ID 30086992) and pMIR2111-3:GUS[11,32]) and ecotype MG-20 (wild type and tml-1[33]). Generation of stable transgenic plants expressing pUBQ1:MIR2111-3[34] and MIR2111-3 knockout lines followed a published procedure based on callus regeneration[11,35] and made use of Agrobacterium tumefaciens AGL1 and L. japonicus ecotype Gifu B-129.

Further, A. thaliana Col-0 wild type, holt-1 (line ID SALK_044075.49.80.x) and holt-2 (line ID SALK_140092.27.55.x) were used. Generation of p35s::MiR2111b expressing plants as well as pMIR2111a/b:GUS lines was done via floral dipping using A. tumefaciens GV3101. Cloning approaches made use of E. coli strains TOP10 or DB3.1. Plants were infected with M. loti MAFF303099[36] expressing DsRED bacteria.

### Construct generation
For p35s driven miRNA overexpression, the transcription start site upstream of the MIR2111b stemloop was predicted using the publicly

available Softberry toolset with standard settings (http://www.softberry.com/berry.phtml?topic=tssplant&group=programs&subgroup=promoter). The entire precursor gene, including 80–100 bp downstream of the miRNA stem loop, was amplified from *A. thaliana* Col-0 genomic DNA using primers carrying overhangs for subsequent cloning into the gateway vector PGWB602. For the *pMIR2111a/b:GUS* lines, a three kb region upstream of the precited stem loop was cloned into the vector PMDC163 using gateway cloning. For Golden Gate technology-based generation of CRISPR/Cas9 constructs targeting the *MIR2111-3* locus a codon optimized Cas9 endonuclease from the *Streptococcus pyogenes* containing the potato IV2 intron driven by a minimal 35 s promotor was used. Two gRNAs GGTAATCTGCATCCTG and GAGTCGGTATATATTGGGTC were predicted using CLC Main Workbench 8 (Qiagen). Primers can be found in Supplementary Table 2.

## Lotus plant growth

For plant growth and phenotyping, *L. japonicus* seeds were surface scarified, sterilized using sodium hypochloride solution containing 1 g/l NaClO, imbibed in ddH$_2$O and transferred to sterile ¼-strength B&D medium[37] with 1% (w/v) phyto agar (Duchefa Biochemie). Following stratification for three days at 4 °C, seeds were germinated at 21 °C in constant darkness for two (MG-20) or three (Gifu B-129) days. For growth on plates, seedlings were transferred to 12 × 12 cm square plastic dishes containing 50 ml ¼-strength B&D / 1% (w/v) phyto agar medium supplemented with KNO$_3$ at indicated concentrations. Plants were grown at long day conditions (16 h light, 21 °C / 8 h dark, 17 °C). Roots were shaded from direct light. For root architecture evaluation and quantification of molecular miR2111 and *TML* levels, plants were grown for twoweeks. For quantification of lateral root initiations plants were grown for ten days.

## Arabidopsis plant growth

For plant growth and phenotyping, *A. thaliana* seeds were sterilized by 30 minutes incubation in a solution of 70% (v/v) ethanol and 0.05% (v/v) Triton X-100. Sterile seeds were transferred to 12×12 cm square plastic dishes containing 50 ml ½-strength MS medium[38] without nitrogen / 1% (w/v) phyto agar medium supplemented with KNO$_3$ at indicated concentrations and stratified for three days at 4 °C. Plants were grown at long day conditions (16 h light, 21 °C / 8 h dark, 17 °C). Roots were shaded from direct light. For quantification of molecular miR2111 and *HOLT* levels, plants were grown for ten days. For quantification of lateral root initiations plants were grown for seven days.

## Lotus grafting

Plants were treated and germinated as described above. After germination plants were transferred to ¼-strength B&D / 1 mM KNO$_3$ / 1% (w/v) phyto agar medium. Plates were kept in darkness for two days, then eight days in long day conditions. For graftings, seedlings were cut near the lower end of the hypocotyl, and immediately submerged in water. New shoots were transplanted onto root stocks and arrested using silicone tubing (Ø 0.64 mm). Grafted plants were transferred to fresh medium and covered with filter paper soaked in ddH$_2$O, then grown at long day conditions for two to three weeks. Prior to phenotyping or tissue harvest, tubing was removed from chimeric plants to determine grafting success. Lotus hetero-grafting involving two distinct ecotypes followed a different procedure.

## Lotus hetero grafting

Plants were treated and germinated as described above. After germination plants were transferred to ¼-strength B&D / 1 mM KNO$_3$ / 1% (w/v) phyto agar medium. Plates were kept three days in long day conditions. For graftings, seedlings were cut near the middle of the hypocotyl, and immediately submerged in water. New shoots were transplanted onto root stocks and arrested using silicone tubing (Ø

0.5 mm, ~3 mm long). Grafted plants were transferred to fresh medium, then kept at 26 °C 22 h light for five days to enable graft site regeneration. Afterwards grafted plants were incubated for two more weeks at long day conditions (16 h light, 21 °C / 8 h dark, 17 °C). Prior to phenotyping or tissue harvest, tubing was removed from chimeric plants to determine grafting success.

## Arabidopsis grafting

Plants were treated and germinated as described above (section 'Arabidopsis plant growth'), and grown for five days on ½-strength MS / 1% (w/v) phyto agar medium medium with full nitrogen content[38]. Grafting followed a published protocol, utilizing sterile precision forceps and a sapphire blade[39]. For grafting one cotyledon of Arabidopsis plants was removed and cut at the hypocotyl. Cut Arabidopsis plants were reassembled on a sterile nitrocellulose membrane on sterile water soaked Whatman paper. After grafting, plants left to recover at 26 °C for four days, then grown for seven more days on ½-strength MS medium containing 100 μM KNO$_3$ at long day conditions before evaluation.

## Split root assay

Plants were treated and germinated as described above (section 'Lotus plant growth'). After germination, root tips were cut off and plants transferred to ¼-strength B&D/0.5 mM KNO$_3$/1% (w/v) phyto agar medium. After 10 days at long day conditions, plants which had generated two secondary roots were selected for onward processing and transferred to plates containing slices of ¼-strength B&D/1.5% (w/v) phyto agar medium. Plants were positioned in a way that secondary roots were placed on separate agar patches not in physical contact with each other and containing KNO$_3$ concentrations as indicated. Plants were grown for 13 more days under long day conditions until phenotypic evaluation or tissue collection.

## Root phenotypic analysis

Plate grown plants were scanned using a conventional scanning system (CanoScan 8800 F, Canon). Image processing made use of OpenCV (https://opencv.org/) functions. Root architectural traits were measured using an in-house Python script. The script was optimized to recognize Lotus roots by color contrast and relied on manual confirmation.

## RNA extraction and quantitative PCR (qRT-PCR) assays

For total RNA extractions, plant or aphid tissue was shock frozen in liquid N. Total RNA was extracted by a modified Lithium Chloride-TRIzol LS (ThermoFisher) protocol[40]. Plant RNA was extracted from tissue of at least ten independent plants per biological replicate. RNA was eluted in DEPC-treated water, RNA concentration was determined using a Nanodrop device (ThermoFisher). RNA was DNAse treated using DNAseI (ThermoFisher) according to manufacturer guidelines. cDNA was prepared using SuperScriptIV (ThermoFisher) or RevertAid (ThermoFisher) reverse transcriptase following a previously optimized pulsed protocol[11,41]. Briefly, RNA and primers (2 μM odT, 0,5 μM specific primers) were mixed and incubated for 5 min at 65 °C. The remaining reaction mix was assembled and incubated at 16 °C for 30 mins followed by 60 cycles (30 °C for 30 s, 42 °C for 30 s and 50 °C for 1 s) and 5 min at 85 °C for enzyme inactivation. Stemloop primers for reverse transcription of small RNAs were designed such that the six basepairs at the 5' end of the stemloop primer were complementary to six nucleotides at the 3' end of the small RNA, for reverse transcription of Lotus mRNAs oligo dT primer was used, for *A. thaliana* RNA only locus specific primers were used for reverse transcription (Supplementary Table 3). The RT-reaction was assembled according to manufacturer's guidelines using 500 ng of total RNA. qRT-PCRs were assembled using SensiFAST™ SYBR® No-ROX mastermix (Bioline) at 10 μl reaction size and 500 nM primer concentration. Levels of target genes were

normalized to levels of two independent reference genes, Lotus *ATP SYNTHASE2* and Lotus *PROTEIN PHOSPHATASE2a* or Arabidopsis *PROTEIN PHOSPHATASE2a* and Arabidopsis *UBIQUITIN EXTENSION PROTEIN 2* or U6 (Supplementary Fig. 11a). qRT-PCR reactions were executed in a BioRad CFX384 lightcycler (BioRad). Primers are listed in Supplementary Table 4. Data analysis made use of LinRegPCR[42].

### Aphids

For aphid experiments, we used *Planococcus citri*[43], which could be propagated on *L. japonicus* as sole host plant. Lotus plants were infected with aphids by placing an infested host stem with a small aphid population onto young, four week-old plants growing in a 3:1 clay granule (2–5 mm, Lamstedt): vermiculite (3–6 mm, Isola Vermiculite GmbH) mixture saturated with ¼-strength B&D medium. After two weeks, aphids were collected and snap frozen in liquid nitrogen. RNA extraction, as well as DNAse treatment, cDNA synthesis and qRT-PCRs, were performed as described. For qRT-PCR experiments on aphid RNA extracts we used aphid α-Tubulin as normalization reference[44].

### Staining and microscopic analysis

GUS staining and fixation was performed as described[11]. Plants were fixed by incubation in 1x phosphate buffer (50 mM $NaH_2PO_4$, 50 mM $Na_2HPO_4$, pH 7.0) supplemented with 4% Paraformaldehyde, followed by 3 washing steps using 1x phosphate buffer. For GUS staining, samples were incubated overnight at 37 °C in X-Gluc buffer (0.5 mg/ml X-Gluc, 1 mM $K_4(Fe(CN)_6)$, 1 mM $K_3(Fe(CN)_6)$, 0,05% Triton X-100 in 1x phosphate buffer). After 3 more washing steps in 1x phosphate buffer, plants were incubated in a buffer containing acetic acid, glycerol and ethanol (ratio 1:1:3) at 60 °C until tissue was cleared from chlorophyll.

Whole plant phenotypes were monitored, and photographs taken using a Leica MZ FLIII stereomicroscope. For analysis of semithin sections, fixed, GUS-stained roots or leaves were further dehydrated and embedded in resin (Kulzer Technovit 7100). Sections were prepared using a Leica RM2065 microtome, then analyzed and documented using a Zeiss Imager M2 microscope. For quantification of lateral root initiations, roots were separated from shoots and fixed using 4% Paraformaldehyde in 1x PBS buffer, cleared using ClearSee (10% Xylitol, 15% sodium deoxycholate and 25% Urea in $ddH_2O$) and stained with Fluorol Yellow (0.01% in 96% Ethanol) as described[45]. Stained roots were scanned using a Leica SP8 confocal microscope, and images were used for phenotypic analysis. Fluorol yellow was imaged at $λ = 488$ nm excitation and $λ = 520–588$ nm emission, and additionally, transmission white light was observed. The resulting integrated images of whole roots allowed quantification of lateral root primordia at early, pre-emergence stages. Lateral root initiations include pre-emergence primordia as well as emerged lateral roots irrespective of the developmental stage (Supplementary Fig. 2).

### Data analysis and graphical representation

Data analysis made use of Python 3.7.x using the libraries Statsmodels and Pandas. Plots were generated with the python libraries Matplotlib and Seaborn. Boxplot center lines show the medians, outer box limits indicate the 25th and 75th percentiles. Whiskers extend 1.5 times the interquartile range from the 25th and 75th percentiles, or the last data point. Data points are represented as dot. Dotplots center lines indicate the average value of all data points. All statistical tests used were two-sided. For pairwise comparison we used *t*-tests for multi-comparison we used ANOVA and post hoc Tukey-HSD testing. Results of ANOVA or t-test analyses, biological replicate numbers and individual datapoints are listed in the Source Data file.

### Reporting summary

Further information on research design is available in the Nature Portfolio Reporting Summary linked to this article.

## Data availability

All primary data and images analysed in the context of this study are available from the corresponding author upon request. Source data are provided with this paper.

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

## Acknowledgements

We thank Johanna Schröter and the ZMBP gardening team for dedicated plant care, Caterina Brancato for help with plant transformations, Dugald Reid for help with construct generation, Angela Fischer for designing Supplementary Figure 1b, and Laura Ragni for discussions. Our thanks further go to Christoph Weiste, Wolfgang Dröge-Laser and Arthur Korte for critical reading of the manuscript. We apologize to authors whose work could not be cited due to space limitations. This research was supported by the German Research Foundation (grant CRC1101, project C07), Ministry of Science, Research and Art of Baden-Wuerttemberg (Az:7533-30-20/1).

## Author contributions

M.Se., H.B., M.Sch., E.R., C.W. and U.H. performed experiments and analyzed data; K.M., M.Se. and H.B. conceived and designed research; K.M. and M.Se. wrote the paper.

## Funding

## Competing interests

The authors declare no competing interests.
