## [Peer Review File · Nature Communications]

A micro RNA mediates shoot control of root branchingReviewer #1 (Remarks to the Author):

In the manuscript, Sexauer et al. report the new role of miR2111-TML regulon, which was previously identified as an important part of a systemic root-shoot-root regulatory mechanism called autoregulation of nodulation (AON). They show that miR2111-TML has a role of regulating lateral roots formation. Interestingly, the authors extend their finding in *Lotus* to non-nodulating plants of *Arabidopsis* and show that the miR2111-TML regulon is basically conserved in the two species with a different responsive mechanism to nitrate. The research topic in the manuscript is interesting in terms of systemic shoot control of root branching. Experiments are mostly well done. However, as I mention below, current data do not sufficiently support the authors conclusion and there are several issues that need to be addressed by additional experiments.

Major:

Given the phenotypic data set, it is likely that miR2111-TML regulon is involved in root branching in the two species. The potential underlying mechanism is mostly unclear. Also, although it is mentioned in the manuscript that root architecture is related to nutrient acquisition, no data is presented to show that miR2111-TML regulon is indeed related to nutrient acquisition. These should be addressed by additional experiments, which may include nitrate uptake assay and gene expression of nitrate transporter genes. It is also interesting to see the verification of genetic dependency/independency of CEP and cytokinin pathways.

The analysis of holt is based on single allele. Complementation test and/or other mutant phenotypic are required to conclude the role of HOLT in root branching.

It is unclear what mutations in miR2111 is occurring in miR2111ko plants. The indel information in the CRISPR plants should be presented. It is particularly important to state that the ko plants produce no miR2111.

As the root branching of Gifu are unaffected by nitrate, all data set (WT, miR2111ox, miR2111ko, tml) should be provided using MG-20 background. Some important data are missing.

Minor:

Primary root formation may be also presented to discuss root architecture.

The name HOLT is a little confusing. How about simply AtTML?

Phylogenetic analysis is helpful to know the relationship between TML and HOLT.

miR2111 expression is repressed by rhizobia inoculation. Is this related to lateral root formation?

In *Arabidopsis*, miR2111 is reported to be induced by phosphate starvation. The connection of the phosphate-responsive expression and root branching may be at least discussed.

It is interesting to see if other players of AON such as HAR1 and CLE-RS are also involved in the regulation of root branching.

Reviewer #2 (Remarks to the Author):

In this manuscript Sexauer et al., investigated the role of miR2111 in lateral root initiation and N responsiveness using miR2111 overexpressing and knock-out lines, and knockouts of the miR2111 target gene TML and its orthologue HOLT in *Lotus japonicus* and *Arabidopsis thaliana*, respectively. They found that grafting miR2111 overexpressing shoots onto wild type (wt) rootstocks resulted in reduced number of lateral root (LR) initiation in *L. japonicus* (Figure 2d) and *A. thaliana* (Figure 4i), and the roots of the chimeric plants accumulated higher level of miR2111 and lower level of TML transcripts (Figure 1 d,f; A.t.) when compared to control (wt/wt) grafts. From these experiments the authors concluded that miR2111 acts as systemic mobile regulator of lateral root initiation across dicot plant lineages. However, the authors did not test the mobility of the

miRNA2111 precursor from the shoot to the root. Thus, it cannot be excluded that the miRNA precursors are mobile, which then get processed into mature miRNAs in the recipient root tissue. This could have been performed by labelling the overexpressed miRNA precursors with single nucleotide polymorphism (SNP) and then monitoring the accumulation of miRNA precursors with and without SNPs in the grafted rootstocks. More importantly, to unequivocally establish the role of miR2111 in systemic control of lateral root initiation, the miRNA2111 knockout lines should be used as scions (shoots) in the grafting experiments under increasing concentrations of nitrate in the media.

Based on Extended data – Methods, only single time points (developmental stages) were analysed. In the experiments involving *L. japonicus*, 10- and 14-days old seedlings were used for investigating lateral root initiation and for evaluating root architecture and gene expression levels, respectively. In the experiments involving *A. thaliana*, 7- and 10-days old seedlings were used for quantifying lateral root initiation and for analysing gene expression levels, respectively. This limits the understanding of the role of miR2111 in LR initiation in the context of root development.

Additional comments:

Figures are not self-explanatory, and the figure legends are rather poor. For example, no (or limited) information is provided about the tissues used for assessing the levels of miR2111 and TML/HOLT, and which part of the grafts was analysed. What does G stand for in wt(G)?

What do the authors mean by relative level? How was the normalization performed? Did the authors establish that the expression level of the Lotus ATP SYNTHASE2 and PROTEIN PHOSPHATASE2a, and the Arabidopsis PROTEIN PHOSPHATASE2a and UBIQUITIN EXTENSION PROTEIN 2 does not change during root development and under different nitrate levels?

More than one independent miR2111 overexpressing lines should be included in the grafting experiments to add weight to the authors' claims.

No information was provided about the CRISPR/Cas9 construct used to make miR2111-3 knock-outs (cloning, vector, promoter and terminator sequences etc) and about the analysis of the miR2111-3 knock-outs.

Why was tml-5 excluded from subsequent experiments such as Fig 2c? It highlights the lack of independent biological repeats.

Nitrogen levels used in experiments Fig1 and 2 should be indicated to link and correlate the data with Fig3. (Not just Fig 2b)

Why does Fig 3a use tml-5 and tml-6 and Fig 3c only tml-1? Fig 3b, why is the 5 mM data point missing?

Fig 4d, it is not known how many independent ox mutant plants were used in these experiments and how many times the experiments were repeated.

REVIEWER COMMENTS AND AUTHOR RESPONSES

NCOMMS-22-02574-T, A micro RNA mediates shoot control of root branching

Reviewer #1 (Remarks to the Author):

In the manuscript, Sexauer et al. report the new role of miR2111-TML regulon, which was previously identified as an important part of a systemic root-shoot-root regulatory mechanism called autoregulation of nodulation (AON). They show that miR2111-TML has a role of regulating lateral roots formation. Interestingly, the authors extend their finding in Lotus to non-nodulating plants of Arabidopsis and show that the miR2111-TML regulon is basically conserved in the two species with a different responsive mechanism to nitrate. The research topic in the manuscript is interesting in terms of systemic shoot control of root branching. Experiments are mostly well done. However, as I mention below, current data do not sufficiently support the authors conclusion and there are several issues that need to be addressed by additional experiments.

Major:

1.1 Given the phenotypic data set, it is likely that miR2111-TML regulon is involved in root branching in the two species. The potential **underlying mechanism** is mostly unclear. Also, although it is mentioned in the manuscript that **root architecture is related to nutrient acquisition**, no data is presented to show that miR2111-TML regulon is indeed related to nutrient acquisition. **These** should be addressed by **additional experiments**, which **may include nitrate uptake assay and gene expression of nitrate transporter genes**.

Author response:

We agree with the reviewer that our data does not support a direct involvement of the miR2111-TML regulon in nitrate acquisition, nor is this part of our hypothesis. We don't, as the reviewer implies, suggest or expect the miR2111-TML regulon to have a direct impact on nitrate acquisition via regulation of nitrate transporter genes (*NRTs*). Following the reviewers' request, we have, at the example of *A. thaliana*, tested whether *HOLT* loss in *holt-1* mutants or *MIR2111b* overexpression has an impact on *NRT* transcript abundance levels. qRT-PCR analysis did not reveal significant differences in *NRT1.1*, *1.5*, *2.1* or *3.1* expression in roots of these lines compared to wild type plants (Supplementary Figure 13a-d, revised manuscript). To avoid misunderstandings, we have revised the manuscript text to ensure that formulations that may suggest an implication of miR2111-TML regulon in nitrate uptake are avoided.

Our data does, however, suggest a role of the miR2111-TML regulon in root branching in response to substrate nitrate levels. Lateral root number, or degree of branching, directly affect the root surface area. As nitrate uptake by roots takes place through its contact zone with the rhizosphere, an increased root surface is hypothesized to represent a pre-condition for enhanced nitrate uptake (lines 265-272, revised manuscript).

1.2 It is also interesting to see the verification of genetic dependency/independency of CEP and cytokinin pathways.

Author response:

Regulation of miR2111 and *TML* in response to rhizobial infection depends on the cytokinin receptor *LHK1* (Tsikou et al., 2018, Fig. 3)¹ which is involved in infection-induced cytokinin signalling and induction of root-induced CLE-RS3 peptides (Tsikou et al. 2018, Fig. S8D)¹. *HAR1*, which was shown to interact with another CLE-peptide (CLE-RS2) and is assumed to act as universal shoot receptor of root-derived CLE peptides in the context of rhizobial infection², was similarly required for restricting miR2111 abundance in response to rhizobia (Tsikou et al., 2018, Fig. 3)¹. To meet the reviewers request, we have tested whether nitrate-induced miR2111 regulation similarly relies on CLE-peptide mediated cytokinin signalling through *LHK1* and *HAR1*. Interestingly, nitrate-dependent downregulation of miR2111 was not dependent on functional *LHK1* or *HAR1* (rebuttal, Figure 1).

Similarly, the CEP peptide receptor CRA2 was shown to be required for rhizobia-induced miR2111 abundance control (Gautrat et al., 2020, Fig. 3)³, whereas nitrate-induced abundance control of miR2111 is wild type-like in *cra2* mutants (Gautrat et al., 2020, Fig. 4B)³.

In summary, we think that there is a nitrate-specific mechanism regulating miR2111 expression independent of rhizobia-induced miR2111 regulation. Deciphering this mechanism will be an interesting endeavour, but exceeds the scope of this manuscript. We thus prefer not to include these data in the present manuscript, as we think it would distract from its main message.

Figure 1 Nitrate dependent miR2111 regulation is independent of *L. japonicus* (Lotus) *LHK1* and *HARI*. Relative levels of mature *miR2111* in shoot tissue of Lotus ecotype Gifu, *har1-3*, *lhk1-1* and *har1-1 lhk1-1*. qRT-PCR analyses. RNA levels are relative to those of two reference genes. Comparison used Student's t-test (*= $p \leq 0.05$, **= $p \leq 0.01$).

1.3 The analysis of holt is based on single allele. Complementation test and/or other mutant phenotype are required to conclude the role of HOLT in root branching.

Author response:

As suggested by the reviewer, we have included a second *holt* allele, *holt-2*, in the experiments. This line is phenotypically indistinguishable from *holt-1*, and a foraging response, as observed in wild type plants, is absent in both *holt-1* and *holt-2* (Fig. 4f, revised manuscript).

1.4 It is unclear what mutations in miR2111 is occurring in miR2111ko plants. The indel information in the CRISPR plants should be presented. It is particularly important to state that the ko plants produce no miR2111.

Author response:

The *MIR2111-3* knockout line bears a 12 bp deletion in the stem-loop region of the locus, immediately adjacent to the miR2111a sequence. In the revised manuscript, we include a figure showing the deletion as well as its predicted effect on the secondary structure of the stemloop transcript (Supplementary Fig. 3, revised manuscript). While the *MIR2111-3* knockout line still produces miR2111, presumably from other *MIR2111* loci, mature miR2111 levels are significantly reduced, and *TML* mRNA levels are increased in *MIR2111-3* knock out plants (Supplementary Fig. 4a,b, revised manuscript).

1.5 As the root branching of Gifu are unaffected by nitrate, all data set (WT, miR2111ox, miR2111ko, tml) should be provided using MG-20 background. Some important data are missing.

Authors:

The generation of additional miR2111 overexpression or miR2111 knockout plants in a second ecotype would have been very time-consuming, as the procedure requires regeneration from callus tissue, and *L. japonicus* displays a long generation time. To meet the reviewers request, we thus conducted heterografting experiments, where we grafted ecotype Gifu scions onto ecotype MG20 root stocks (Supplementary Fig. 6b). The Gifu (shoot)/MG20 (root) control exhibited a nitrate responsive lateral root initiation phenotype similar to non-grafted plants of the ecotype MG20. In contrast, miR2111ox (shoot; ecotype Gifu)/MG20 (root) grafts show no nitrate responsiveness and a reduced LR initiation number compared to wild type Gifu (shoot)/MG20 (root) grafts. This demonstrates that independent of the respective root ecotype, *MIR2111-3* knockout shoots resulted in a loss of nitrate responsiveness in lateral root initiation numbers.

Minor:

1.6 Primary root formation may be also presented to discuss root architecture.

Author response:

We comparatively investigated primary root growth in *tml* and *holt* mutants, but did not identify differences compared to wild type plants of the respective species (rebuttal, Fig. 2). In order not to expand the manuscript unnecessarily, we thus chose not to discuss primary root formation in the manuscript, as it does not appear to grant further insights.

Figure 2 Primary root length is independent of *TML/HOLT*. **a** Primary root length of *A. thaliana* Col-0 compared to *holt-1* 10 days post germination at 1 mM nitrate. **b**, primary root length of *L. japonicus* ecotype MG20 wild type compared to *tml-1* 14 days post transfer at different nitrate concentrations. Comparisons used Student's t-test (n.s., not significant, where $p \geq 0.05$) (**a**) or analysis of variance (ANOVA) and post-hoc Tukey testing ($p \leq 0.05$) (**b**), with distinct letters indicating significant differences.

1.7 The name HOLT is a little confusing. How about simply AtTML?

Author response:

The name '*TML*' had previously been assigned to a different gene in Arabidopsis (AT5G57460.1, *Tplate Complex Muniscin-Like*⁴). We therefore sought to identify a name that has not been pre-assigned in this species but still offers a direct reference to the legume name of the gene, *TML*^{5,6}. With this in mind, *HOLT* (*Homolog Of Legume TML*) seemed appropriate to us.

1.8 Phylogenetic analysis is helpful to know the relationship between TML and HOLT.

Author response:

The phylogenetic relationship between *TML* homologs in dicot plant lineages including legumes and Brassicaceae has been analysed and discussed previously^{1,6}. To take reference to this point, we refer to

these publications in the revised manuscript, but avoid including a similar analysis to avoid redundancies.

1.9 miR2111 expression is repressed by rhizobia inoculation. Is this related to lateral root formation?

Author response:

Quantification of lateral root initiations in plants inoculated with rhizobia compared to mock-treated plants demonstrated that rhizobial inoculation indeed results in a reduction of lateral root initiations. Interestingly, this effect was independent of *TML*, suggesting that rhizobial inoculation impacts root primordium formation in a *TML*-independent manner. We have included the respective dataset in the revised manuscript (Supplementary Fig. 7).

1.10 In Arabidopsis, miR2111 is reported to be induced by phosphate starvation. The connection of the phosphate-responsive expression and root branching may be at least discussed.

Author response:

We included a short discussion about phosphate responsiveness of miR2111 in the revised manuscript (lines 273-275).

1.11 It is interesting to see if other players of AON such as HAR1 and CLE-RS are also involved in the regulation of root branching.

Author response:

Please see our response to comment 1.2.

Reviewer #2 (Remarks to the Author):

In this manuscript Sexauer et al., investigated the role of miR2111 in lateral root initiation and N responsiveness using miR2111 overexpressing and knock-out lines, and knockouts of the miR2111 target gene TML and its orthologue HOLT in *Lotus japonicus* and *Arabidopsis thaliana*, respectively. They found that grafting miR2111 overexpressing shoots onto wild type (wt) rootstocks resulted in reduced number of lateral root (LR) initiation in *L. japonicus* (Figure 2d) and *A. thaliana* (Figure 4i), and the roots of the chimeric plants accumulated higher level of miR2111 and lower level of TML transcripts (Figure 1 d,f; A.t.) when compared to control (wt/wt) grafts. From these experiments the authors concluded that miR2111 acts as systemic mobile regulator of lateral root initiation across dicot plant lineages.

2.1 However, the authors did not test the mobility of the miRNA2111 precursor from the shoot to the root. Thus, it cannot be excluded that the miRNA precursors are mobile, which then get processed into mature miRNAs the recipient root tissue. This could have been performed by labelling the overexpressed miRNA precursors with single nucleotide polymorphism (SNP) and then monitoring the accumulation of miRNA precursors with and without SNPs in the grafted rootstocks

Author response:

To meet this concern, we conducted aphid experiments, where we allowed aphids (*Phenacoccus solenopsis*) to feed on *L. japonicus* phloem sap. Using stemloop qRT-PCR, we specifically detected mature miR2111 in cDNA prepared from aphid RNA (Supplementary Fig. 1c, revised manuscript). As this assay does not detect unprocessed precursor transcripts, this strongly suggests mature miR2111 as mobile element either as a duplex or single strand. These observations are in line with a previous report suggesting that mobile miRNAs undergo vascular movement as mature duplexes⁷.

2.2 More importantly, to unequivocally establish the role of miR2111 in systemic control of lateral root initiation, the miRNA2111 knockout lines should be used as scions (shoots) in the grafting experiments under increasing concentrations of nitrate in the media.

Author response:

As *MIR2111-3* expression was only traceable in aboveground plant parts¹, grafting of the knockout lines seems unnecessary to establish the role of shoot miR2111 on lateral root formation under varying KNO_3 conditions. To meet this request, we thus opted to characterise the nitrate response in non-grafted *MIR2111-3* knockout plants, as grafting would induce additional variation. Under starvation conditions, numbers of lateral root primordia of *MIR2111-3* knockout plants showed no difference compared to wild type plants (Supplementary Fig. 6c, revised manuscript). Interestingly, *MIR2111-3* knockout plants displayed a significant nitrate-dependent increase in lateral root initiations that was absent in wild type plants (Supplementary Fig. 6c, revised manuscript), indicating a role of shoot miR2111 in suppressing lateral root primordium formation under high nitrate supply.

2.3 Based on Extended data – Methods, only single time points (developmental stages) were analysed. In the experiments involving *L. japonicus*, 10- and 14-days old seedlings were used for investigating lateral root initiation and for evaluating root architecture and gene expression levels, respectively. In the experiments involving *A. thaliana*, 7- and 10-days old seedlings were used for quantifying lateral root initiation and for analysing gene expression levels, respectively. This limits the understanding of the role of miR2111 in LR initiation in the context of root development.

Author response:

To trace the role of the miR2111-*TML* node across developmental stages, we used *L. japonicus* ecotype MG20 wild type and *tml-1* plants as an example (Supplementary Fig. 14, revised manuscript). Across a timeframe of 21 days, the comparison revealed an increasing *tml*-dependent difference in root growth between the two lines.

In the remainder of the manuscript we focused on earlier timepoints mainly for practical reasons, as older roots are significantly more difficult to analyse reliably with respect to root architectural features.

Additional comments:

2.4 Figures are not self-explanatory, and the figure legends are rather poor. For example, no (or limited) information is provided about the tissues used for assessing the levels of miR2111 and TML/HOLT, and which part of the grafts was analysed. What does G stand for in wt(G)?

Author response:

As suggested, we have thoroughly checked all figure legends and included additional information where appropriate to improve accessibility and readability.

2.5 What do the authors mean by relative level? How was the normalization performed? Did the authors establish that the expression level of the Lotus ATP SYNTHASE2 and PROTEIN PHOSPHATASE2a, and the Arabidopsis PROTEIN PHOSPHATASE2a and UBIQUITIN EXTENSION PROTEIN 2 does not change during root development and under different nitrate levels?

Author response:

In our qRT-PCR assays, we make use of a relative quantification approach. To minimize risks of bias due to possible variation in reference gene expression levels, we have used two independent standard reference genes for normalization in both model species (*Lotus japonicus*: ATP Synthase2 and Protein Phosphatase2 (PP2A); *Arabidopsis thaliana*: PP2A and Ubiquitin Extension Protein2). Neither of these genes were found to show altered expression levels under varying nitrate supply^{8,9}.

As described in the methods section, and following standard procedures, we have co-amplified reference genes from each cDNA sample to normalize RNA/cDNA content. Target gene abundances were first determined as relative values to either reference gene. The arithmetic mean of these two values is used to generate graphs. To improve accessibility, we have set the mean of datapoints for one of the conditions presented in each graph as '1', and related the remaining mean expression values to this. This does not change the data or conclusions, it merely improves readability, and is common practice.

2.6 No information was provided about the CRISPR/Cas9 construct used to make miR2111-3 knock-outs (cloning, vector, promoter and terminator sequences etc) and about the analysis of the miR2111-3 knock-outs.

Author response:

The *MIR2111-3* knockout line displays a 12 bp deletion in the stem-loop region of the *MIR2111-3* transcript. In the revised manuscript, we included a visualization demonstrating the position and extension of the deletion (Supplementary Fig. 3, revised manuscript), and included details on the CRISPR/Cas9 construct generation in the cloning section (Supplementary Information, lines 25-29, revised manuscript).

Further, we included a molecular characterization of miR2111 and *TML* transcript levels in the *MIR2111-3* knockout line (Supplementary Figure 4, revised manuscript). The line shows significantly reduced miR2111 levels paralleled by increased *TML* levels compared to wild type plants (supplementary Figure 4). The residual miR2111 population it produces may be derived from loci other than *MIR2111-3*¹⁰.

2.7 Nitrogen levels used in experiments Fig1 and 2 should be indicated to link and correlate the data with Fig3. (Not just Fig 2b)

Author response:

We improved the figure captions and indicated the respective nitrate conditions used for tissue generation in all depicted experiments.

2.8 Why was *tml-5* excluded from subsequent experiments such as Fig 2c? It highlights the lack of independent biological repeats.

Why does Fig 3a use *tml-5* and *tml-6* and Fig 3c only *tml-1*?

Author response:

In this manuscript we make use of two different *L. japonicus* ecotypes, Gifu and MG20. Accordingly, the mutant lines we use have different ecotype backgrounds:

MG20: *tml-1*

Gifu: *tml-5*, *tml-6*, 2111ok, 2111ko

The two ecotypes show different growth behaviours and nitrate responses (Fig. 3, Supplementary Fig. 5a, Supplementary Fig. 6, revised manuscript). We thus displayed mutants and their respective wild type controls of each ecotype in separate graphs for better comparability.

Our data show that in both ecotypes, nitrate responsiveness of lateral root initiation is miR2111 dependent (Fig. 3, Supplementary Fig. 5a, Supplementary Fig. 6, revised manuscript). For ecotype Gifu, as *tml-5* and *tml-6* lines behaved alike, we focused our experiments on one of the alleles, *tml-6*, for practical reasons. In our view, our use of multiple *tml/holt* mutant alleles in different ecotypes as well as different species, paired with the molecular data we present are sufficient to back the conclusions we draw.

All experiments shown in this manuscript are based on at least two and up to six independent biological replicates. The number of biological replicates used in each experiment is listed in Supplementary Table 3.

2.9 Fig 3b, why is the 5 mM data point missing?

Author response:

We included the 5 mM data point in Fig. 3b.

2.10 More than one independent miR2111 overexpressing lines should be included in the grafting experiments to add weight to the authors' claims.

Author response:

In the revised manuscript, we include a characterisation of three independent miR2111 overexpression lines, which show similar phenotype and overall strong expression of miR2111 (Supplementary Fig. 11 a-c, revised manuscript). In our view, there is no further insight to be expected from using more than one of these lines in grafting experiments. As these are extremely time consuming, we have thus refrained from setting up additional replicates using different lines.

2.11 Fig 4d, it is not known how many independent ox mutant plants were used in these experiments and how many times the experiments were repeated.

Author response:

The data shown in Figure 4d-f consist of 3 independent replicates. The experiment was repeated 3 more times without the *holt-2* allele showing the same results. Per RNA extraction, tissue of at least 10 independent plants was used for each biological replicate. This information has been added to the respective methods section. The number of biological replicates represented in each Figure panel is specified in Supplementary Table 3 (revised manuscript). In Fig. 4 d-f (revised manuscript).

References

- 1 Tsikou, D. *et al.* Systemic control of legume susceptibility to rhizobial infection by a mobile microRNA. *Science* **362**, 233-236 (2018). <https://doi.org:10.1126/science.aat6907>
- 2 Okamoto, S., Shinohara, H., Mori, T., Matsubayashi, Y. & Kawaguchi, M. Root-derived CLE glycopeptides control nodulation by direct binding to HAR1 receptor kinase. *Nat Commun* **4**, 2191 (2013). <https://doi.org:10.1038/ncomms3191>
- 3 Gautrat, P., Laffont, C. & Frugier, F. *Compact Root Architecture 2* promotes root competence for nodulation through the miR2111 systemic effector. *Curr Biol* **30**, 1339-1345 e1333 (2020). <https://doi.org:10.1016/j.cub.2020.01.084>
- 4 Kleffmann, T. *et al.* The Arabidopsis thaliana chloroplast proteome reveals pathway abundance and novel protein functions. *Curr Biol* **14**, 354-362 (2004). <https://doi.org:10.1016/j.cub.2004.02.039>
- 5 Magori, S. *et al.* *Too Much Love*, a root regulator associated with the long-distance control of nodulation in *Lotus japonicus*. *Mol Plant Microbe Interact* **22**, 259-268 (2009). <https://doi.org:10.1094/MPMI-22-3-0259>
- 6 Takahara, M. *et al.* *Too Much Love*, a novel Kelch repeat-containing F-box protein, functions in the long-distance regulation of the legume-*Rhizobium* symbiosis. *Plant Cell Physiol* **54**, 433-447 (2013). <https://doi.org:10.1093/pcp/pct022>
- 7 Devers, E. A. *et al.* Movement and differential consumption of short interfering RNA duplexes underlie mobile RNA interference. *Nat Plants* **6**, 789-799 (2020). <https://doi.org:10.1038/s41477-020-0687-2>
- 8 Høgslund, N. *et al.* Dissection of symbiosis and organ development by integrated transcriptome analysis of *Lotus japonicus* mutant and wild-type plants. *PLoS One* **4**, e6556 (2009). <https://doi.org:10.1371/journal.pone.0006556>
- 9 Vidal, E. A. *et al.* Integrated RNA-seq and sRNA-seq analysis identifies novel nitrate-responsive genes in Arabidopsis thaliana roots. *BMC Genomics* **14**, 701 (2013). <https://doi.org:10.1186/1471-2164-14-701>
- 10 Okuma, N., Soyano, T., Suzaki, T. & Kawaguchi, M. *MIR2111-5* locus and shoot-accumulated mature miR2111 systemically enhance nodulation depending on *HAR1* in *Lotus japonicus*. *Nat Commun* **11**, 5192 (2020). <https://doi.org:10.1038/s41467-020-19037-9>

Reviewer #1 (Remarks to the Author):

The revised manuscript has addressed most of the concerns I have noted. The data support the authors' claims. The following minor comments may need to be further considered.

Supplementary Fig. 2 may need to be revised, for example, the number of cortex layers should correspond to Lotus.

The term, "ko" (knockout), should be carefully used. It is likely that the 2111ko plants used in the analysis has a reduced function of miR2111-3, but sufficient evidence has not been presented to be considered that it is a knockout. The authors did not provide data showing the amount of miR2111 derived from the miR2111-3 locus.

Reviewer #2 (Remarks to the Author):

The revised manuscript has shown considerable improvement. However, a crucial experiment is still missing to support the authors' claim that shoot-derived mobile miR2111 controls lateral root initiation. As indicated in the original review, 'To unequivocally establish the role of miR2111 in the systemic control of lateral root initiation, the miRNA2111 knockout lines should be used as scions (shoots) in grafting experiments.' In the rebuttal, the authors stated that 'As MIR2111-3 expression was only traceable in aboveground plant parts, grafting of the knockout lines seems unnecessary to establish the role of shoot miR2111 in lateral root formation.' However, miR2111 is also expressed in both shoot and root vasculature, as demonstrated by GUS staining of five-day-old stably transformed *A. thaliana* seedlings expressing pMIR2111a:GUS and pMIR2111b:GUS (Supplementary Fig. 10 and Fig. 4a). Hence, the aforementioned experiment is necessary to substantiate the authors' claim. It is surprising that Sexauer et al. only used miR2111 overexpressing lines in the corresponding grafting experiments (Fig. 1 and Supplementary Fig. 6).

REVIEWER COMMENTS AND AUTHOR RESPONSES

NCOMMS-22-02574A, A micro RNA mediates shoot control of root branching

Reviewer #1 (Remarks to the Author):

The revised manuscript has addressed most of the concerns I have noted. The data support the authors' claims. The following minor comments may need to be further considered.

Supplementary Fig. 2 may need to be revised, for example, the number of cortex layers should correspond to Lotus.

Author response:

Irrespective of the plant species concerned, we intended Supplementary Fig. 2 as a sketch illustrating the terminological difference between lateral root primordia, emerged lateral roots and lateral root initiations. To address the reviewers' concern, we further simplified the figure to ensure a species-independent validity of the sketch.

The term, “ko” (knockout), should be carefully used. It is likely that the 2111ko plants used in the analysis has a reduced function of miR2111-3, but sufficient evidence has not been presented to be considered that it is a knockout. The authors did not provide data showing the amount of miR2111 derived from the miR2111-3 locus.

Author response:

Thank you for this comment. In the revised manuscript, we refer to the line as *mir2111-3-1*. *MIR2111-3* is a polycistronic locus generating the mature isoforms miR2111a and miR2111b. Both mature isoforms are also produced by other *MIR2111* loci. As a result, we are unable to associate mature miR2111 sequences with their locus of origin.

Reviewer #2 (Remarks to the Author):

The revised manuscript has shown considerable improvement. However, a crucial experiment is still missing to support the authors' claim that shoot-derived mobile miR2111 controls lateral root initiation. As indicated in the original review, 'To unequivocally establish the role of miR2111 in the systemic control of lateral root initiation, the miRNA2111 knockout lines should be used as scions (shoots) in grafting experiments.' In the rebuttal, the authors stated that 'As *MIR2111-3* expression was only traceable in aboveground plant parts, grafting of the knockout lines seems unnecessary to establish the role of shoot miR2111 in lateral root formation.' However, miR2111 is also expressed in both shoot and root vasculature, as demonstrated by GUS staining of five-day-old stably transformed *A. thaliana* seedlings expressing pMIR2111a:GUS and pMIR2111b:GUS (Supplementary Fig. 10 and Fig. 4a). Hence, the aforementioned experiment is necessary to substantiate the authors' claim. It is surprising that Sexauer et al. only used miR2111 overexpressing lines in the corresponding grafting experiments (Fig. 1 and Supplementary Fig. 6).

Author response:

As the activity of the *L. japonicus MIR2111-3* locus is restricted to the phloem of leaf veins (Tsikou et al., 2018; see also Figure 2g, revised manuscript), we did not consider grafting experiments using *mir2111-3-1* plants as necessary to support our conclusions. We thus refrained from performing these experiments in the first revision, also because the experimental setup is technically demanding and time consuming.

In response to the renewed reviewers' comment we now added the respective experiments (Figure 2f).